# Improving the Prognostic and Predictive Value of Circulating Tumor Cell Enumeration: Is Longitudinal Monitoring the Answer?

**DOI:** 10.3390/ijms251910612

**Published:** 2024-10-02

**Authors:** Anna Fabisiewicz, Malgorzata Szostakowska-Rodzos, Ewa A. Grzybowska

**Affiliations:** Department of Molecular and Translational Oncology, Maria Sklodowska-Curie National Research Institute of Oncology, Roentgena 5, 02-781 Warsaw, Poland; anna.fabisiewicz@nio.gov.pl (A.F.); malgorzata.szostakowska-rodzos@nio.gov.pl (M.S.-R.)

**Keywords:** circulating tumor cells, longitudinal analysis, CTC dynamics

## Abstract

Circulating tumor cell (CTC) numbers in the blood of cancer patients can indicate the progression and invasiveness of tumors, and their prognostic and predictive value has been repeatedly demonstrated. However, the standard baseline CTC count at the beginning of treatment, while informative, is not completely reliable and may not adequately reflect the state of the disease. A growing number of studies indicate that the long-term monitoring of CTC numbers in the same patient provides more comprehensive prognostic data and should be incorporated into clinical practice, as a factor that contributes to therapeutic decisions. This review describes the current status of CTC enumeration as a prognostic and predictive factor, highlights the shortcomings of current solutions, and advocates for longitudinal CTC analysis as a more effective method of the evaluation of developing disease, treatment efficacy, and the long term-monitoring of the minimal residual disease. As evidenced by the described reports, the longitudinal monitoring of CTCs should provide a better and more sensitive prediction of the course of the disease, and its incorporation in clinical practice should be beneficial.

## 1. Introduction

CTC enumeration is a well-documented prognostic and predictive biomarker in cancer management, but its implementation in clinical practice is still ongoing and encounters technical, logistic, and conceptual problems. Apart from the fact that CTC detection techniques need standardization and improvement, CTC enumeration is quite expensive, and while not very aggravating for patients, it requires skills and resources to meet the standards of clinical diagnostics. Improving its prognostic and predictive value would give clinicians a more reliable tool, useful for therapeutic decisions. While technical improvements in CTC detection, classification, and enumeration is one avenue of research that leads to this goal, another one might be the longitudinal monitoring of a patient. As shown in this review, data from serial blood collections are more reliable than only one, baseline evaluation, and enable us to track the progress of the disease.

## 2. The Current Status of CTC Research

### 2.1. Old and New Blood-Based Cancer Biomarkers for Screening and Progression

Tumor biomarkers have critical value in cancer screening, early diagnosis, recurrence detection, and therapeutic monitoring. They are diverse and can be detected in blood, tissue samples, and biofluids (saliva, feces, urine). There are several known blood-derived tumor biomarkers, including tumor-specific antigens—PSA (prostate-specific antigen), SCCA (squamous cell carcinoma antigen), AFP (alpha-fetoprotein specific to hepatocellular carcinoma), CA125 (glycosylated mucin specific mostly to ovarian cancer), CA19-9 (carbohydrate antigen 19-9, pancreatic and colorectal cancer), and LDH (lactate dehydrogenase, melanoma, renal and colorectal cancer)—and less specific ones such as CEA (carcinoembryonic antigen) and CA 15-3 (carbohydrate antigen 15-3, mucin) (see reviews on tumor biomarkers [1,2]). Some biomarkers, such as, for example, CEA and CA15-3 in breast cancer, are associated with recurrence and metastasis [3]. However, the efficacy of these classic biomarkers is limited [4,5,6], and recently, novel biomarkers have been introduced, including circulating tumor DNA (ctDNA) and circulating tumor cells. These new biomarkers, although still undergoing clinical trials or preclinical studies, have great potential for becoming a significant part of clinical practice and can improve the efficiency of treatment [7,8]. Important advantages of these biomarkers lie in their minimal invasiveness and relative safety, especially compared to a real biopsy, because they only require the collection of blood samples. There are several possible applications of CTC-based information in clinics: the screening and early detection of tumors, the early indication of recurrence in postoperative patients, and the monitoring of advanced disease. Molecular data acquired during CTC analysis (e.g., expression profiling or mutational analysis) can be used for the identification of new therapeutic targets [9,10,11,12,13], identifying the mechanisms of resistance [14,15,16,17,18,19], heterogeneity [20,21], and, in general, tailoring the therapeutic response to a specific patient, paving the way to personalized medicine.

The clinical importance and implications of CTC studies for basic research have been already extensively reviewed [22,23,24,25], so in this review, we focus on the role of longitudinal monitoring as a way to improve the clinical performance of CTC analysis.

### 2.2. Biology and the Role of CTCs in the Metastatic Process

CTCs are extremely rare and difficult to observe, so despite the fact that they were first noticed in 1869 [26], most of our knowledge about them has been gathered in the last two decades. Due to the current development of the techniques of CTC detection, we are able to not only detect and enumerate but also isolate and characterize CTCs from the blood of patients.

CTCs can be classified as single CTCs and CTC clusters, including homotypic clusters composed of tumor cells and heterotypic clusters with immune cells [27]. Several reports indicated that CTC clusters are more metastatic [28,29,30], but because of their rarity, there are only a few reports on their clinical significance [31,32].

Many reports and clinical trials have demonstrated that CTC numbers are predictive and prognostic (described in Section 2.3), but we still need to gain a deep understanding of the importance of CTC dynamics and the correlation between CTC status and disease progression. Our current understanding of the metastatic cascade assigns the role of ‘metastatic seeds’ to CTCs, but this is true only for a very small fraction of these cells (0.01%, [33,34]). The fact that a high number of CTCs in advanced metastatic disease is prognostic and predictive is probably due to the simple fact that progressing metastatic lesions produce and—proportionally—shed more cells, but it may also signify an additional dissemination and seeding of new metastatic sites, which contributes to metastatic burden. In postoperative disease, the increase in numbers may reflect the fact that the disseminated cells started proliferation, creating a true macro-metastatic lesion, which is associated with shedding some of these cells into the bloodstream. In both cases, the close monitoring of CTC numbers should act as a warning sign preceding any clinical manifestation, and its detection should help in therapeutic decisions.

### 2.3. Molecular Characteristics of CTCs

Many reports focused on the molecular characteristics of isolated CTCs as a way to determine specific changes in these cells toward a more metastatic phenotype, especially compared to primary tumor cells. If included in standard clinical practice, this may be a step towards personalized medicine. The characterized features included specific mutations [35,36,37], the EMT score [38,39,40], the expression of EpCAM and stem cell markers [41,42], DNA methylation dynamics [43,44,45], and the whole expression profile [46,47,48].

Several reports highlight the discrepancies between the genetic status of the primary tumor and CTCs; some characterize specific activating mutations that occur in CTCs, which are not present in the primary tumor, for example, in luminal breast cancer, in which mutations in the ESR1 and PIK3CA genes are associated with resistance to hormonal therapy [49,50]. Mutational studies based on ctDNA probably have a better clinical perspective than those based on CTCs, because of a simpler procedure with similar significance [51,52,53].

From the accumulated studies on genotypic and phenotypic variability in CTCs, we can conclude that CTCs are very heterogeneous, even within the same patient [37,42,54]. This may reflect the initial heterogeneity of the primary tumor or changes that occur in secondary lesions, since in the metastatic setting, the detected cells/DNA are derived from metastases.

A molecular analysis of CTCs is important from the perspective of basic scientific knowledge but also potentially holds clinical utility; further analyses of CTC biology, heterogeneity, and phenotypic plasticity should demonstrate the feasibility of including it in clinical evaluation.

### 2.4. Clinical Significance of CTC Enumeration and Profiling

Based on the accumulated evidence on the importance of CTCs as a prognostic marker in many cancers, researchers study the utility of CTC enumeration in clinical settings. Several multicenter prospective clinical trials were performed, but only some ended with definitive positive results that may be helpful in the clinic. Currently, worldwide clinical guidelines do not consider the use of CTCs (except the inclusion of CTCs for cM0 classification); however, gained data predict their great potential in clinical applications.

Most data on the presence of CTCs were obtained in the best investigative studies of breast cancer (BC). In a pooled analysis of 3173 patients with localized BC from five centers, with the cut-off set at ≥1 CTCs/7.5 mL of blood, the CTC count was identified as an independent prognostic factor for disease-free survival (DFS) and overall survival (OS) [55].

In another large trial in 1697 early BC patients, a benefit of radiotherapy in an adjuvant setting for patients with detectable CTCs was shown [56]. The association between CTC presence in neoadjuvant settings (preneoadjuvant chemotherapy and pre-surgery) and OS or DSF has been reported in many recent studies [57]. Furthermore, CTCs detected in minimal residual disease (MRD) after the removal of the primary tumor or ending of neoadjuvant treatment, when clinical parameters show a lack of active disease, may earlier predict recurrences [55]. 

In metastatic breast cancer (MBC), ≥5 CTCs was set as being correlated with poor survival [58]. The development of the disease and the response to therapy can be monitored by the CTC number. Cristofanilli et al. showed that CTC levels and the dynamics of CTCs over time help to identify high-risk MBC patients. In a retrospective study of 2436 MBC patients from 18 centers, they demonstrated that CTC numbers are useful for the stratification of patients with advanced disease. Stage IV indolent patients (<5CTC) had a longer OS than those in stage IV aggressive (36.3 months vs. 16 months), independently of the disease subtype [59].

Numerous clinical trials were conducted to address the issue of CTCs that serve as a biomarker; in some of these trials, CTCs were only enumerated (SWOG S0500, [60] STIC CTC, [61] whereas in others, their phenotype was assessed (CirCe T-DM1 trial [62], DETECT study, [63]). These trials and other ongoing phase III and phase II clinical trials are overviewed and summarized in the reviews by Lin et al. [7] and Stoecklein et al. [64].

In many solid cancers, an analogous correlation between CTC number and poor prognosis has been observed. The prognostic value of CTCs for OS in localized colorectal cancer was observed in some studies and confirmed in a meta-analysis that included 3687 patients from 20 centers [65]. Another meta-analysis performed in 1329 patients with metastatic colorectal cancer (CRC) showed that OS (overall survival) and PFS (progression free survival) decreased in patients with CTC [66]. The same conclusion was drawn from other studies and meta-analyses [67,68]. All data obtained in early and advanced CRC suggested a worse prognosis in the presence of CTCs. 

The same observations were made for lung cancer [69,70,71,72], prostate cancer [73], melanoma [74], and head and neck cancer [75]. 

The risk of recurrence correlates with the number of CTCs; more CTCs indicate a higher risk of recurrence, while a decrease in or clearance of CTCs correlates with good therapeutic response. However, in the phase III SUCCESS study, approximately 88% of patients with a high number of CTCs did not show recurrence and metastatic progression after 36 months of follow-up [76]. The patients in this trial were treated with chemotherapy depending on the presence of CTCs. This result highlights the problem of the overtreatment of many patients who may unnecessarily suffer side effects from redundant chemotherapy. Inversely, some patients with undetectable or a low number of CTCs will develop distant metastasis. Therefore, there is a great need for a deeper understanding of CTC characteristics and additional biomarkers.

The other problem is that some studies have shown that changes in CTC number during therapy do not correspond to response criteria defined in the RECIST (Response Evaluation Criteria in Solid Tumors) guidelines [77]. Therefore, CTC enumeration has not been included in the RECIST guidelines yet.

### 2.5. The Shortcomings of a Single, Non-Recurrent CTC Analysis

Despite all the efforts and achievements presented in the previous chapter, the clinical utility of CTCs is still hindered by inconsistencies, from technical difficulties in standardization to more theoretical questions about the reliability of the zero CTC count in a single blood draw. These issues should be addressed if the CTC count is to become a reliable clinical tool.

#### 2.5.1. The Methodology of CTC Detection Requires Standardization

Although CTC research has made a huge technological leap in the last two decades, there are still unsolved issues due to inconsistencies in protocols and differences in various detection systems. The two FDA-approved systems, CellSearch and Parsortix, differ in the principle of detection, which inevitably leads to the characterization of different subpopulations of CTCs, depending on the method.

Most of the results of CTC enumeration are obtained with CellSearch [78], the first FDA-approved system (in January 2004) for the detection and monitoring of CTCs in patients. The CellSearch method is based on immunoaffinity to EpCAM (Epithelial Cell Adhesion Molecule), and it captures only cells with epithelial features, potentially omitting EpCAM-low and negative CTCs. The system is used for both research and diagnostics, and while relatively expensive, its price is decreasing. With more standardization and the acceptance of its diagnostic role, it should become affordable for clinics.

The second most popular system for the capture and harvest of CTCs—Parsortix PC1—was approved by the FDA in May 2022. The Parsortix system uses a microfluidic technology enabling the epitope-independent capture of all CTC phenotypes based on cell size and deformability, allowing for CTC enumeration and subsequent downstream analysis [79]. This system is currently being evaluated in many studies and clinical trials, as reviewed in Wishart et al. [80].

EpCAM-dependent enrichment was found to be more specific and suitable for clinical applications than size-dependent, label-free approaches, but the latter were found to be more suitable for molecular characterization [81]. The heterogeneity of CTC results in significant variations in surface biomarker expression [41,82], so the uniform recognition of all types of CTCs by labeling-dependent methods using identical standard is impossible. Size-dependent or image-based approaches include subpopulations that may be lost in the EpCAM-dependent CellSearch system, namely cells after EMT, clusters, and CTCs masked by other types of cells. However, it has also been reported that since CTCs are fragile, some methods of detection, particularly size-based methods, may damage them [83]. 

Additionally, there are many other systems, label-dependent, size-dependent, or image-based systems, that do not have FDA approval and are used only for research (reviewed in [83,84,85]). In conclusion, it is necessary to establish maximally standardized protocols and address the problem of the group of cancers in which CTCs do not express EpCAM. 

#### 2.5.2. Fluctuations in CTC Numbers Depend on Circadian Rhythm, Clinico-Pathological Features, and Therapeutic Interventions

The release of CTCs from tumors is not constant and may differ not only between different patients with comparable disease status but also within the same patient at different time points. Moreover, it was demonstrated that CTCs can clear within minutes after release into the bloodstream [86], so the timing of the blood draw can be crucial. Several factors have been considered to promote the release of CTCs. Donato et al. [87] demonstrated that the intravasation of highly metastatic CTC clusters is caused by hypoxia. There are also reports suggesting that CTC release may be regulated by the circadian rhythm and accelerates during the rest phase [88,89,90].

Several reports highlight the possibility that CTC shedding may be associated with specific treatment [91]. Surgical intervention was shown to increase the number of CTCs [92,93,94,95]. This effect may confuse the statistical significance of the results and should be considered for baseline time selection. Pang et al. [96] observed that CTCs measured up to the seventh day after surgery did not affect PFS and OS, while CTCs measured on day 14 after surgery were related to PFS. They concluded that the time point of the CTC measurement affects the prognostic value.

An increase in CTC number has been also reported after radiation therapy [97] and after needle biopsy [98,99], prompting the question of whether the mobilization of CTCs during cancer therapy may cause metastasis [100]. These data suggest that therapeutic intervention may inadvertently contribute to tumor cell dissemination, which could be counterproductive. This prompts the question of whether we should address this problem on the clinical level by, for example, introducing some perioperative course of cytotoxic or other CTC-targeting treatment?

The concerns of therapy-induced CTC mobilization were also expressed with respect to chemotherapy, although, due to the variability in the available treatments, these responses are more complex. Ito et al. [101] reported CTC mobilization after chemotherapy in a xenograft mouse model of human pancreatic cancer. Ortiz-Otero et al. [102] reported chemotherapy-induced CTC release in metastatic cancer patients with a spectrum of cancer types. 

On the other hand, many reports have shown that CTC numbers decrease after treatment. For example, Vetter et al. [103] demonstrated that the denosumab treatment of metastatic BC significantly reduced the number of CTCs in patients’ blood. Denosumab is a monoclonal antibody against the receptor activator of the nuclear factor-κB ligand (RANKL). It suppresses osteoclasts and prevents bone resorption, mitigating the effects of breast cancer bone metastasis. 

Bendahl et al. [104] also observed the decline in CTC numbers in small cell lung cancer, after standard chemotherapy, and Lozano et al. [105] observed the same effect upon docetaxel treatment in metastatic castration-resistant prostate cancer (mCRPC) patients. In this last case, early CTC decline after treatment was demonstrated to be a better predictor of survival than PSA.

The decline in CTC numbers after therapy is usually interpreted as a sign of the efficacy of the specific treatment, which advocates for using this parameter for therapy monitoring. Numerous reports indicate that a persistently high CTC count during chemotherapy is prognostic, as well as low/no CTC numbers, which is a favorable treatment response (see Section 3).

## 3. Principles and Advantages of Longitudinal Analysis

To address some of the problems and challenges described in the previous chapter, Cristofanilli et al. [58] in an early study from 2004 on MBC attempted a more systematic approach and included a follow-up blood collection in addition to the baseline. They observed that CTC monitoring during treatment provided additional information. This was confirmed by the pooled study by Bidard et al. [106], in which the authors reported that CTC changes during treatment are significant for PFS and OS and indicated that tracking CTC changes improves prognostic accuracy. This observation was followed by more articles with the baseline and the follow-up count, exploring the question of the clinical value of the follow-up count [60,107]. As a natural consequence, some other teams followed the obvious path of exploring whether more blood collections are even more informative and clinically useful.

### 3.1. The Serial Monitoring of CTC Dynamics in a Single Patient Is Prognostically Superior to Single or Baseline/Follow-Up Blood Collection

CTC reports with more than two blood collections are still not very frequent, since they require much more work, coordination, and cooperation with the clinic. Despite the greater workload and the level of complication, the number of longitudinal reports is increasing, and they repeatedly demonstrate the usefulness of longitudinal studies. While most reports focus on CTC numbers, some reports include a molecular analysis of isolated CTCs, which enables us to trace genetic and phenotypic changes in some cancer-specific markers.

Gerratana et al. [108] comparatively investigated the CTC count in a longitudinal setting and ctDNA analysis for MBC monitoring. The CTCs were analyzed in 74 patients, with three blood collections: baseline, evaluation, and progression. The authors observed an increase in CTC numbers only at progression. They also expressed an interesting opinion that while both CTC enumeration and ctDNA characterization provided prognostic information, CTC numbers are more related to the underlying metastatic biology, whereas ctDNA provides a more quantitative assessment of tumor burden. 

Szostakowska-Rodzos et al. [109] studied the dynamics of CTC numbers in a longitudinal study with three subsequent blood collections (every 3 months) from 135 MBC patients. Statistical analysis demonstrated that constantly increasing CTC numbers are unfavorable, while a constantly low (<5) CTC count is a strong favorable predictor for PFS and OS. These results indicate that the dynamics of CTC changes provide additional clinically important information.

Some reports focus not only on single CTCs but also take into account the numbers and prognostic value of CTC clusters. Larssen et al. [31] in a prospective observational trial analyzed CTCs/CTC clusters from 156 patients with MBC. They analyzed CTC numbers from the baseline and the three subsequent blood collections (after 1, 3, and 6 months). The results indicate that the prognostic value of CTC count ≥ 5 CTCs and CTC cluster evaluation increases over time, suggesting that the dynamic of CTCs and CTC clusters are more relevant to prognosis than a single baseline enumeration and that the presence of CTC clusters adds to prognostic value. Wang et al. [32] came to the same conclusions in their longitudinal study of MBC in which they also analyzed CTC clusters. They also observed that a larger-size CTC cluster conferred a higher risk of death.

Longitudinal analysis not only provides information concerning changes in CTC numbers in the same patient, but it also allows for some additional analyses; for example, it enables us to trace the evolution of the status of the selected markers. Forsare et al. [110] comparatively studied estrogen receptor (ER) status in primary tumors (PTs) and CTCs in 147 breast cancer patients. ER expression was evaluated in CTCs at baseline and after 1 and 3 months of endocrine therapy. The authors demonstrated a shift from ER positivity to negativity between PTs and CTCs at different time points. They proposed that the ER positivity of CTCs reflects the retention of a favorable phenotype that still responds to therapy, and as such, it can be clinically useful.

Phenotypic plasticity was also observed by Cohen et al. [111]. The authors used a microcavity array platform (size-based approach) to count and characterize CTCs from 184 MBC patients at up to nine time points at 3-month intervals. Taking advantage of their approach, without bias towards epithelial characteristics, they observed that a shift from an epithelial to a mesenchymal expression pattern in the isolated CTCs is associated with inferior clinical outcomes.

Stergiopoulou et al. [112] reported the molecular characterization of isolated CTCs as a tool for the early detection of minimal residual disease (MRD) in breast cancer. The study described long-term CTC monitoring (up to 126 months) with a very comprehensive analysis of isolated CTCs (enumeration, phenotypic analysis, gene expression analysis, mutation analysis in CTCs and the corresponding plasma ctDNA, DNA methylation analysis) performed in 13 patients. The authors concluded that this type of analysis can reveal the presence of MRD 4 years before the appearance of clinically detectable metastasis. This proves that the molecular characterization of CTCs holds huge clinical potential.

Although breast cancer is the most studied, other types of cancer are also explored. For example, Hendricks et al. [113] presented a prospective pilot study in which they monitored a cohort of 44 colorectal cancer (CRC) patients of miscellaneous tumor stages with as many as five blood collections in addition to the baseline. The study analyzed the complicated kinetics of CTCs in CRC patients after the resection of the primary tumor and provided data concerning the CTC quantity over a long-term follow-up. 

In the pursuit for a novel, more sensitive biomarker for MRD, Ko et al. [114] performed a longitudinal analysis of CTCs (with four time points) in 21 metastatic patients with nasopharyngeal carcinoma (NPC). While limited with a small sample size and short follow-up time, this study also indicated the association of CTC numbers with PFS and demonstrated that CTC detection could be a more sensitive tool for tracking MRD than standard methods (plasma EBV assay, PET-CT).

### 3.2. Clinical Relevance of Long Monitoring in Relation to Treatment

Usually, blood collection time points are designed to match significant clinical factors, such as the beginning of a new treatment or a scheduled clinical or radiological evaluation. This coordination of CTC enumeration with clinical procedures enables us to evaluate the response to therapy. Establishing CTC count as an early predictor of progression and as a tool for the systematic monitoring of the treatment efficacy is one of the main goals of this research. Consequently, several reports addressed the issue of longitudinal monitoring and its clinical value for ongoing treatment. The reports published so far indicate that CTC monitoring can be used as a complementary tool for clinical evaluation as a basis for determining treatment efficacy and treatment choices.

Pang et al. [96] addressed the question of the optimal time points for CTC analysis in a study of 110 breast cancer patients in relation to treatment. Their study revealed that the most significant correlation between CTC numbers and PFS and OS was observed for the baseline and the end point of follow-up collection, while CTCs detected before chemotherapy were only related to PFS. Furthermore, the numbers of CTCs at the last adjuvant chemotherapy were more correlated with prognosis than those before adjuvant chemotherapy. As mentioned before, they also observed postoperative variability in CTC numbers, which indicates that this time point is unreliable.

Another recent study on MBC by Magbanua et al. [115] was carried out with three or more serial blood draws from 469 patients for CTC enumeration. In this study, the data collected from patients were divided into three models used for PFS and OS end point estimation. The data from baseline blood collection (bCTC) in which blood was drawn at the new cycle of chemotherapy were compared with the data at the end of the chemotherapy cycle to estimate the change in CTCs (cCTC). Survival analysis showed that patients negative for CTC at both time points had a significantly higher median PFS than patients with CTCs in any or all time points. Moreover, the data from all collections were used for the CTC trajectory (tCTC) model development, which enabled us to estimate the trend in CTC numbers during treatment: high, mid, low, and negative. The results of this study showed that the patients with negative tCTC (without CTCs) had a better prognosis for PFS and OS and that patients with high tCTC had the worst prognosis for PFS and OS.

In another study from the same group, Magbanua et al. [116] evaluated the prognostic and predictive value of CTC monitoring in hormone receptor-positive metastatic breast cancer. Patients were randomized to letrozole alone or letrozole plus bevacizumab in the first-line setting. The authors studied data from 294 patients at the baseline and in the three subsequent blood collections (before every third bevacizumab cycle and approximately 21-day intervals in the letrozole-only arm). They found that CTCs were highly prognostic for the addition of bevacizumab to first-line letrozole. 

A study by Bendahl et al. [104] explored the prognostic value of CTC presence in small cell lung cancer (SCLC) and assessed the dynamics of CTCs in longitudinal samples from a patient cohort from the RASTEN phase III trial, a randomized controlled study designed to estimate the survival benefit of the addition of low-molecular-weight heparin (LMWH) to standard chemotherapy. Baseline blood collection and two other blood collections were performed in 42 patients with limited and extensive disease (even subgroups, 50%). The study demonstrated that the persistent presence of CTCs during and after completion offers additional prognostic information in addition to baseline CTCs. 

A study by Ko et al. [117] (from the same group as in the previously described study of MRD in NPC) provided data of the serial monitoring of treatment outcomes for locally advanced esophageal squamous cell carcinoma (ESCC). The 88 patients included in this analysis received neoadjuvant treatment and curative resection treatment, and the goal was to establish whether CTC enumeration could be used as a tool for the prediction of treatment efficacies, prognostication, and real-time tracking of MRD for the earlier detection of relapse. The authors determined the clinical usefulness of specific time points of CTC detection, namely the following: pretreatment, post-treatment/pre-surgery, and 1 month and 3 months post-surgery. 

## 4. Conclusions

The longitudinal monitoring of CTCs holds the promise of a better, more sensitive, and earlier prediction of the relapse or development of drug resistance. The serial CTC count in patients during treatment should become a useful tool for monitoring and predicting tumor progression and guide treatment choices. As described in this review, these types of studies (summarized in Table 1) repeatedly demonstrate the usefulness of CTC count in the real-time treatment monitoring and risk stratification strategy, with a persistently high CTC count being a strongly unfavorable factor and low/no CTC count being strongly favorable factors. The incorporation of these assays in clinical practice (as proposed in Figure 1) should be highly beneficial, exciding the usefulness of the single/double CTC count, but they should be incorporated depending on the specific context (type and stage of cancer, type of treatment). A molecular analysis of isolated CTCs also offers potential clinical benefits, providing more insight into the current state and evolution of the disease. The CTC count was also investigated as a tool for the long-term monitoring of MRD. These results form a reliable base for introducing CTC-based information in clinical practice.

The CTC count at the baseline is prognostic and may contribute to the choice of a more or less aggressive therapy, but it cannot provide information about the efficiency of the treatment in a specific patient. During the subsequent evaluations of progression, the CTC count can serve as a complementary factor, with a persistently high/low CTC number as an indicator of the response to treatment.

## Figures and Tables

**Figure 1 ijms-25-10612-f001:**
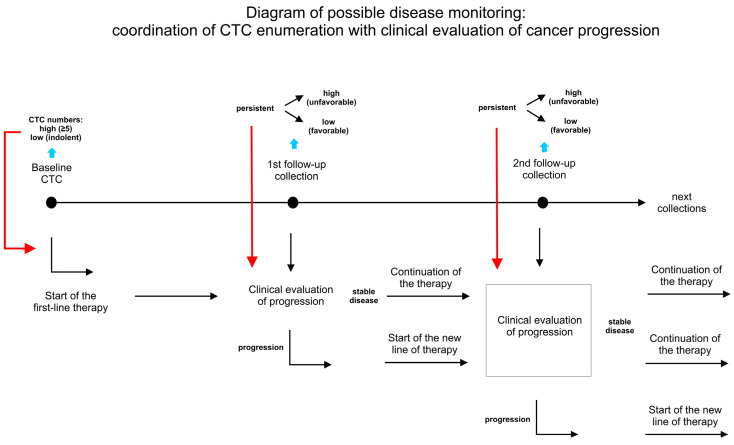
Diagram depicting proposed monitoring of disease and efficiency of therapy with CTC count (red arrows) complementing clinical evaluation.

**Table 1 ijms-25-10612-t001:** A current list of longitudinal CTC studies (with at least 3 blood collections).

Cancer Type	Authors	Year	No. of Patients Included in Analysis	No. of Collections	CTC Detection Method	Main Findings	Ref.
Breast Cancer	Wang C, Mu Z, Chervoneva I, …, Cristofanilli M, Yang, H.	2017	128	3	CellSearch	CTC clusters added additional prognostic values to CTC enumeration alone, and a larger-size CTC cluster conferred a higher risk of death in MBC patients.	[32]
Breast Cancer	Larsson AM, Jansson S, Bendahl PO, …, Rydén L.	2018	152	4	CellSearch	A longitudinal evaluation of CTC and CTC clusters improved prognostication and monitoring in patients with MBC starting first-line systemic therapy. Changes in CTC count throughout treatment were significantly correlated with survival, and the prognostic value was more prominent at later time points. High CTC counts and the presence of clusters were identified as prognostic factors for OS and PFS.	[31]
Breast Cancer	Forsare C, Bendahl PO, Moberg E, …, Rydén L.	2020	147	3	CellSearch	A shift in ER status from PT to DM/CTCs was demonstrated. A retained ER positivity of CTCs after the initiation of systemic therapy was associated with a better prognosis for PFS. This effect was observed only for follow-up samples, highlighting the importance of CTC phenotyping during treatment.	[110]
Nasopharyngeal Carcinoma (NPC)	Ko JMY, Vardhanabhuti VV, Ng WT, …, Lung ML.	2020	21	4	CTChip^®^FR1	CTCs were characterized as a more sensitive biomarker for MRD, when compared with imaging. Longitudinal changes in CTCs and EBV DNA along with CT treatment for mNPC were found to be predictive for disease relapse.	[114]
Breast Cancer	Magbanua MJM, Hendrix LH, Hyslop T, …, Rugo HS.	2021	469	≥3	CellSearch	The authors used the CTC trajectory model, which divided patients into groups, predicting a consistent trend for negative CTCs, low CTCs, mid CTCs, and high CTCs. The mid and high tCTC groups were identified with a higher risk of early progression and shorter PFS and OS.	[115]
Breast Cancer	Magbanua MJM, Savenkov O, Asmus EJ, …, Rugo HS.	2021	294	4	CellSearch	CTC-positive patients at the baseline were identified with a worse PFS and OS than CTC-negative patients. CTC positivity during treatment or baseline was identified as a risk factor for PFS and OS. Patients that became CTC-positive in the 1st follow-up had a poorer prognosis for OS than patients that stayed CTC-negative or patients that remained CTC-positive since baseline. Patients who stayed CTC-positive had a poorer PFS and OS than patients who stayed CTC-negative since baseline.	[116]
Breast Cancer	Gerratana L, Davis AA, Zhang Q, …, Cristofanilli M.	2021	107	3	CellSearch	The ctDNA analysis revealed that mutant allele frequency (MAF) changes followed the response to treatment, while CTC numbers increased only at the time of clinical progression. Conclusion: MAF could be more suitable for real-time disease monitoring, while CTCs could be more likely linked to metastatic biology.	[108]
Breast Cancer	Pang S, Li H, Xu S, …, Zhou G.	2021	164	4	IMNs (immunomagnetic nanospheres)	Surgery led to an increase in the number and prevalence of CTCs on the first day after surgery, and they did not return to the preoperative level until 14 days after surgery. CTC prevalence at the baseline and end point follow-up visits was related to PFS and OS, while the CTCs detected before chemotherapy were only related to PFS.	[96]
Colorectal Cancer	Hendricks A, Dall K, Brandt B, …, Sebens S.	2021	47	5	NYONE, RT-PCR	Surgery did not have any statistically significant effect on the quantity of CTCs detected by the cytological approach utilizing the cell imager NYONE. In one of the patients, a constant increase in CTCs detected via both methods (9 months after the surgery) occurred before the local clinical recurrence (13 months after surgery).	[113]
Breast Cancer	Stergiopoulou D, Markou A, Strati A, …, Lianidou E.	2023	13	≥10	CellSearch	The molecular characteristics of CTCs were highly different even for the same patient at different time points, and they always increased before the clinical relapse. Rapid increases in CTC numbers in months 74 and 122, were associated with metastatic disease documented by biopsy 6 months earlier.	[112]
Breast Cancer	Cohen EN, Jayachandran G, Gao H, …, Reuben JM.	2023	184	9	MCA (microactivity array)	This study reaffirmed the cut-off of ≥5 CTCs for an inferior prognosis of patients with MBC. It also highlighted that epithelial CTC counts were prognostic before the initiation of therapy and early in therapy, whereas a shift towards mesenchymal CTC phenotypes as detected by gene expression was associated with disease progression.	[111]
Esophageal Squamous Cell Carcinoma	Ko JMY, Lam KO, Kwong DLW, …, Lung ML	2023	88	12	CTChip^®^FR1	The changes in CTC status pre-surgery and 1 or 3 months after surgery and pT staging after resection are independent prognostic factors of poor prognosis for locally advanced ESCC patients receiving surgical treatment, as well as the presence of CTC clusters, unfavorable CTC status at baseline, and 1 month and 3 months post-surgery.	[117]
Small Cell Lung Cancer	Bendahl PO, Belting M, Gezelius E.	2023	42	2	CellSearch	CTC presence at the baseline was identified as a poor prognostic factor for the survival of patients. A persistent CTC presence at 2-month follow-up and baseline was associated with a significantly higher HR for OS.	[104]
Breast Cancer	Szostakowska-Rodzos M, Fabisiewicz A, Wakula M, …, Grzybowska EA.	2024	135	3	CytoTrack	A high CTC count was an independent poor prognosis marker for PFS and OS, regardless of the time of enumeration. Consistently low CTCs numbers during treatment were revealed to be favorable prognostic markers for PFS and OS. Rising values of CTC counts were identified as predictors for rapid progression.	[109]

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
