# Peer review of "Improving the Prognostic and Predictive Value of Circulating Tumor Cell Enumeration: Is Longitudinal Monitoring the Answer?"

_ijms, 2024, doi:10.3390/ijms251910612_

Round 1

Reviewer 1 Report

Comments and Suggestions for Authors

This review highlights the complexity of the presence or absence of Circulating Tumor Cells in patients and the difficulty to correlate their number with the clinical status. Indeed, CTCs may be a snapshot of the tumor heterogeneity at the given time.  

This review cites numerous references and stays focus on recent and relevant longitudinal studies (studies including a significant number of patients).

The present manuscript is excellent and this paper gives a very complete overview of the interest of CTCs in patient care. The only very minor point would be to add a discussion about the cost of such technologies. Indeed, only the CellSearch system has been cleared by the FDA and the reimbursement of such analyses has not been approved elsewhere. This aspect should be discussed.

Author Response

Comments:

This review highlights the complexity of the presence or absence of Circulating Tumor Cells in patients and the difficulty to correlate their number with the clinical status. Indeed, CTCs may be a snapshot of the tumor heterogeneity at the given time.

This review cites numerous references and stays focus on recent and relevant longitudinal studies (studies including a significant number of patients).

The present manuscript is excellent and this paper gives a very complete overview of the interest of CTCs in patient care. The only very minor point would be to add a discussion about the cost of such technologies. Indeed, only the CellSearch system has been cleared by the FDA and the reimbursement of such analyses has not been approved elsewhere. This aspect should be discussed.

Response:

The authors are grateful for the Reviewer comments and observations.

A small paragraph has been added briefly discussing costs of CTC research (2.5.1., a chapter discussing methodology), however the authors do not feel competent or informed enough to discuss at length the trends in prices or the policies of reimbursement.

Reviewer 2 Report

Comments and Suggestions for Authors

In the review by Fabisiewicz et al., the authors attempt to provide an overview on the topic of CTCs. Overall, the manuscript is rather vague but could be useful for those approaching CTCs for the first time to understand their role in longitudinal studies.

Here are my comments:

  • If the focus of the review is on longitudinal monitoring, it would be more attractive if the review immediately addressed the reasons why CTCs are useful in this context (minimally invasive, only blood samples required). This is only mentioned after comparing CTCs with older markers, but I believe this point should be discussed right after the introduction.

  • Sections 1 (Introduction) and 2.1-2.3 are quite vague, and I believe there is more literature that could be cited. Specifically, in section 2.1, the main applications of CTCs are described, such as screening, early detection, and others. The authors should briefly mention key studies. Moreover, citations are missing. In line 54, the authors mention that molecular analysis can be useful for identifying therapeutic targets, but there is no reference. Additionally, they do not mention that CTC analysis can be useful in studying tumor heterogeneity. This point is important because heterogeneity is a key factor in drug resistance. The authors could briefly touch on this concept. I recommend the following references: 10.3390/biomedicines9091242; doi.org/10.1016/j.ccell.2020.03.007.

  • The conclusions feel rushed. Since CTCs have different translational significance depending on the setting, it would be helpful to clarify what the promises of longitudinal monitoring are in different contexts.

  • I suggest a thorough review of the manuscript to correct editing errors and issues related to references (e.g., lines 135 and 137). Also, check the abbreviations, as they are not defined from the start (e.g., breast cancer and circulating tumor cell).

Comments on the Quality of English Language

Minor editing required

Author Response

Comments:

In the review by Fabisiewicz et al., the authors attempt to provide an overview on the topic of CTCs. Overall, the manuscript is rather vague but could be useful for those approaching CTCs for the first time to understand their role in longitudinal studies.

Response 1:

The manuscript focuses on longitudinal analysis and does not aspire to descirbe CTC research in its full scope, so it may appear vague in an introductionary part. A paragraph with the clarification of the scope was added, along with citations of many excellent reviews on a subject.

Here are my comments:

    If the focus of the review is on longitudinal monitoring, it would be more attractive if the review immediately addressed the reasons why CTCs are useful in this context (minimally invasive, only blood samples required). This is only mentioned after comparing CTCs with older markers, but I believe this point should be discussed right after the introduction.

Response 2:

The appropriate paragraph was added.

    Sections 1 (Introduction) and 2.1-2.3 are quite vague, and I believe there is more literature that could be cited. Specifically, in section 2.1, the main applications of CTCs are described, such as screening, early detection, and others. The authors should briefly mention key studies. Moreover, citations are missing. In line 54, the authors mention that molecular analysis can be useful for identifying therapeutic targets, but there is no reference.

Response 3:

As described earlier, this part was mostly introductory, but the appropriate references were added (in total 19 citations).

Additionally, they do not mention that CTC analysis can be useful in studying tumor heterogeneity. This point is important because heterogeneity is a key factor in drug resistance. The authors could briefly touch on this concept. I recommend the following references: 10.3390/biomedicines9091242; doi.org/10.1016/j.ccell.2020.03.007.

Response 4:

The heterogeneity was addressed in chapter 2.1 and the references were added, however we'd like to point out that this issue was also briefly discussed in the original manuscript, in chapter 2.3 (Molecular characteristics of CTCs)

    The conclusions feel rushed. Since CTCs have different translational significance depending on the setting, it would be helpful to clarify what the promises of longitudinal monitoring are in different contexts.

Response 5:

The authors fully agree that translational significance is dependent on the context, and the appropriate comment was introduced to the Conclusions. The comment however is short, because: 1) we feel that this section should contain a brief summary, so more detailed analysis would take to much space 2) for the moment there is not enough data to outline what specific promises longitudinal analysis offers in which context; we described the available research, but it is still not enough to formulate definite conclusions or guidelines, even in case of the best studied breast cancer.

    I suggest a thorough review of the manuscript to correct editing errors and issues related to references (e.g., lines 135 and 137). Also, check the abbreviations, as they are not defined from the start (e.g., breast cancer and circulating tumor cell).

Response 6:

The errors were corrected and the abbreviations were defined.

Round 2

Reviewer 2 Report

Comments and Suggestions for Authors

I have no further comments. The manuscript can be accepted